# Morbidity of Transrectal MRI-Fusion Targeted Prostate Biopsy at a Tertiary Referral Academic Centre: An Audit to Guide the Transition to the Transperineal Approach

**DOI:** 10.3390/cancers15245798

**Published:** 2023-12-11

**Authors:** Andrea Alberti, Rossella Nicoletti, Paolo Polverino, Anna Rivetti, Edoardo Dibilio, Giulio Raffaele Resta, Pantelis Makrides, Corso Caneschi, Andrea Cifarelli, Antonino D’Amico, Luisa Moscardi, Mattia Lo Re, Federico Peschiera, Maria Lucia Gallo, Alessio Pecoraro, Arcangelo Sebastianelli, Lorenzo Masieri, Mauro Gacci, Sergio Serni, Riccardo Campi, Francesco Sessa

**Affiliations:** Unit of Urological Robotic Surgery and Renal Transplantation, Careggi University Hospital, 50134 Florence, Italy; andrea.alberti@unifi.it (A.A.); pantelis.makr.2012@gmail.com (P.M.); riccardo.campi@unifi.it (R.C.)

**Keywords:** prostate, biopsy, transrectal, transperineal

## Abstract

**Simple Summary:**

The choice between a transrectal or a transperineal approach for MRI-fusion-targeted prostate biopsy is still the subject of debate within the urologic community and across international Guidelines. Specifically, the European Association of Urology (EAU) guidelines recommend abandoning TRPB in favor of TPPB, due to lower rates of infection and sepsis. In contrast, the American Urological Association (AUA) recommends either approach, citing the absence of prospective RCTs assessing infectious risk. In this scenario, transparent reporting of the morbidity of the transrectal prostate biopsy is crucial to inform patients during shared decision-making and to evaluate the cost-effectiveness of different biopsy approaches from clinical and economic standpoints.

**Abstract:**

Despite transrectal prostate biopsy (TRPB) being still widespread globally, the EAU Guidelines strongly recommend the transperineal approach, due to the reported lower infectious risk. Our study aims to evaluate the impact of a standardized clinical pathway for TRPB on post-operative complications. We prospectively collected data from all patients undergoing mpMRI-targeted TRPB at our Academic Centre from January 2020 to December 2022. All patients followed a standardized, structured multistep pathway. Post-procedural complications were collected and classified according to the Clavien–Dindo (CD) Classification. Among 458 patients, post-procedural adverse events were reported by 203 (44.3%), of which 161 (35.2%) experienced CD grade 1 complications (hematuria [124, 27.1%], hematochezia [22, 4.8%], hematospermia [14, 3.1%], or a combination [20, 4.4%]), and 45 (9.0%) reported CD grade 2 complications (acute urinary retention or hematuria needing catheterization, as well as urinary tract infections, of which 2 cases required hospitalization). No major complications, including sepsis, were observed. At uni- and multivariable analysis, age > 70 years and BMI > 25 kg/ m^2^ for patients were identified as predictors of post-operative complications. The results of our study confirm that TRPB is a safe and cost-effective procedure with a low risk of severe adverse events in experienced hands and following a standardized pathway.

## 1. Introduction

Transrectal prostate biopsy (TRPB) is generally considered a safe procedure, although it is noteworthy that approximately 70% of patients experience one or more complications. The majority are minor, self-limiting, non-infectious complications (such as hematuria, hematospermia, hematochezia, and perineal pain), while the most significant morbidity relates to infectious complications, including urinary tract infections, bacteremia, and sepsis [1,2].

There is a debate regarding a higher rate of post-procedural complications associated with TRPB compared to transperineal prostate biopsy (TPPB). Several studies have suggested that TRPB and TPPB exhibit similar rates of minor adverse events, but the discrepancy becomes evident when examining infectious complications [1]. TRPB appears to carry a higher infective risk due to the passage of the biopsy needle through the rectum, introducing the bacterial flora of the rectal mucosa, most commonly Escherichia coli, into the prostate, urinary system, or bloodstream [3,4,5,6]. In contrast, TPPB has been classified as a ‘clean-contaminated’ procedure since the perineal skin can be easily disinfected, avoiding contact with the rectum and its bacterial flora. TPPB requires prophylactic antibiotics targeted to skin flora and common urinary pathogens (e.g., Cephazolin), thereby minimizing infection rates [6]. Historically, Fluoroquinolones have been the agents of choice for prophylaxis due to their excellent pharmacokinetics in prostatic tissue and low resistance rates. However, with the increase in resistance and the recent decision of the European Commission to ban Fluoroquinolones from antibiotic prophylaxis in urological surgery, alternatives are necessary [7]. In this context, Fosfomycin is an attractive alternative because of its broad-spectrum activity against a wide range of Gram-positive and Gram-negative bacteria. Furthermore, it is highly effective against Escherichia coli, the most common causative pathogen of post-TRPB infectious complications [8,9].

Moreover, non-antibiotic strategies have been explored to potentially influence the infection rate in these procedures. Among these, only the transperineal approach and rectal preparation with Povidone-iodine have shown a significant advantage in reducing infectious complications [4,5,10].

Despite any evidence, TPPB shows some shortcomings that might limit its use in daily practice. These include longer procedural time, training, and financial constraints, as well as the need for higher-grade anesthesia compared to TRPB [11]. Hence, the “TREXIT 2020” initiative [12], which recommends switching from TRPB to TPPB in alignment with the EAU Guidelines [13], presents several logistical challenges, requiring the identification of men at high risk for biopsy-related complications and adapting the management accordingly. On the other hand, unlike the European guidelines, the American Urological Association (AUA) recommends both the transrectal and the transperineal approach, due to a lack of evidence on infectious risk [14]. This leaves the debate open on which is actually the best technique and prompts us to reflect on the possibility that there is still a place for TRPB in the diagnostic algorithm of PCa.

In this setting, our study aimed to evaluate the impact of a standardized clinical pathway for TRPB, focusing on post-operative complications, to assess whether this technique could still be a valuable option for the detection of PCa, or if the data reported in the literature for TPPB are superior to those for TRPB, we should forget about TRPB and abandon it.

## 2. Materials and Methods

After the approval from the Ethical Committee, we prospectively collected data from all patients who underwent mpMRI-targeted TRPB at a single referral Academic Centre from January 2020 to December 2022.

All male patients with suspected PCa (e.g., increased PSA, suspected digital rectal examination [DRE]) who underwent mpMRI and had findings of suspicious lesions (PI-RADS ≥ 3), followed by mpMRI-targeted TRPB at our center, were included in the study. Patients without an mpMRI performed before the procedure, as well as those with a negative mpMRI, who underwent systematic biopsy, were excluded.

All TRPB were performed by an expert Urologist (>50 procedures) under real-time ultrasound guidance using a dedicated, fully integrated mobile fusion imaging platform (KOELIS Trinity^®^ MRI TPUS Biopsy System, Grenoble, France) (Figure 1) (https://koelis.com/koelis-trinity/, accessed on 1 November 2023), using a transrectal end-fire transducer equipped with a reusable sterile needle guide kit. All biopsy samples were interpreted by dedicated uro-oncological pathologists, and the results were reported according to the WHO classification of 2016.

The Biopsy procedure includes (1) mpMRI preparation: on mpMRI images, the borders of the prostatic gland and suspicious lesions marked as Region of Interest (ROI) are defined and contoured using dedicated software; (2) TRUS cartography: using a transrectal end-fire transducer, ultrasound scans of the prostate are obtained, and the anatomical limits of the gland are outlined again. (3) fusion of mpMRI images with 3D ultrasound images, creating a 3D cartography for biopsy guidance: during the procedure, the OBT (Organ-Based Tracking Fusion^®^) technology allows the visualization of core location before sampling and its relationship with the ROI as well as accurate recording of core location after sampling. All areas identified on mpMRI with a score of ≥3 according to the PI-RADS classification were considered as possible “suspicious lesions” and subjected to targeted sampling during TRPB. (4) The sampling of cores on the target areas and subsequent systematic sampling of the prostate gland includes two samples (lateral and medio-lateral) from each anatomical area of the prostate (apex, mid-gland, and base). All biopsies were standardly performed, incorporating both target and systematic cores. All biopsies were performed by experienced urologists in an outpatient setting under local anesthesia. Two biopsies per hour were routinely scheduled and performed.

Following our standardized structured multistep pathway (summarized in Figure 2), all patients received one prophylactic dose of Fosfomycin Trometamol (FMT), 3 g administered 3 h before the procedure. For patients with a history of mechanical heart valve replacement or endocarditis, an additional prophylaxis with Amoxicillin-Clavulanate 2 g was administered. In cases where an allergy to Fosfomycin was reported, prophylaxis with Ciprofloxacin 500 mg was administered 3 h before the biopsy. Patients on anticoagulant/antiplatelet therapy required a specialist consultation for possible temporary suspension or replacement of the therapy several days before and after the procedure, depending on the specific medication. A cleansing enema was self-performed at home by each patient the morning before the biopsy.

Before the biopsy, following the collection of medical history and obtaining written informed consent, local disinfection with Povidone-iodine and anesthesia with Lidocaine/Prilocaine 5% cream (EMLA) was performed.

After the biopsy, patients were required to stay in a discharge room for at least 1 h, waiting for the first spontaneous urination, with subsequent visual urine evaluation by a healthcare professional to rule out any potential gross hematuria, acute urinary retention (AUR), or other adverse events.

After discharge, patients were given a dedicated emergency phone number to address any doubts regarding possible peri-procedural complications in the days following the biopsy or to contact a urologist for other procedure-related reasons.

Antibiotic prophylaxis was completed with a second dose of FMT 3 g (or other antibiotic, depending on eventual allergies or medical conditions) after 24 h.

All pre- and peri-procedural data, including age, comorbidities, PSA levels, DRE, mpMRI features, number of samples, and definitive histopathological results, were collected in a dedicated database. Early post-operative complications (within 30 days from TRPB) were collected by a phone interview and classified according to Clavien–Dindo (CD) Classification [15]. Patients who preferred not to answer our post-procedural interview or were unreachable were considered “lost to follow-up”.

The primary objective of our study was to assess the impact of our standardized clinical pathway for TRPB on patient outcomes, focusing on postprocedural complications. The secondary outcome was to identify potential predictors of complications, to select patients with a higher risk of complications, for whom a different diagnostic strategy may be more suitable.

SPSS software (V28.0.1.0) was employed for statistics. Medians and interquartile ranges (IQR) were reported for continuous variables, while frequencies and proportions were reported for categorical variables, as appropriate. The Chi-square test was used to compare different interventions (*p* ≤ 0.01). Univariable and multivariable logistic regression analyses were performed to assess potential predictors for complications.

## 3. Results

Overall, 499 patients underwent mpMRI-targeted TRPB at our Center from January 2020 to December 2022 and were enrolled in our study. Among them, 41 patients (8.2%) were lost to follow-up and thus were not included in the analytical cohort. Baseline characteristics of our sample are reported in Table 1.

Of the 499 patients who underwent mpMRI-targeted TRPB, 69 (13.8%) were not biopsy-naive: 25 (36.2%) were following an active surveillance protocol due to a previous finding of low-risk PCa, while the remaining 44 (63.8%) had persistently elevated PSA values or clinical/imaging progression despite one or more previous negative prostate biopsies. The median total number of biopsy cores taken during the procedure was 16 (IQR: 14–18), with a median of 4 (IQR: 4–6) and 11 (IQR: 10–12) cores for target and random samplings, respectively.

When evaluating post-procedural outcomes, 255/458 (55.7%) patients reported no complication after TRPB, while 203 (44.3%) experienced at least one post-procedural adverse event, most of them classified as Clavien–Dindo (CD) grade 1 (161/203, 79.3%). A detailed summary of all complications is provided in Figure 3.

Among the 203 men who reported complications, 180 (39.3%) had post-TRPB bleeding episodes, including mild short-lasting (median length 3 days, IQR: 2–6) self-resolved hematuria (124/458, 27%) or moderate hematuria (3/458, 0.7%) requiring the placement of a bladder catheter (kept in place for a median of 5 days, IQR: 4.5–6); hematochezia (22/458, 4.8%) with a median length of 2 days (IQR: 1–4); hematospermia (14/458, 3.1%) for a median of 12.5 days (IQR: 5.5–20); or a combination of them (hematuria + hematospermia/hematochezia) (20/458, 4.4%) (a median duration of 7 days; IQR: 4–11). There was no need for blood transfusions, and only one patient accessed the emergency room for evaluation and catheter placement the day after the procedure, while all other cases were managed by healthcare personnel before discharge, by telephone in case of mild symptoms, or in an outpatient setting.

Twenty patients (4.4%) were evaluated for Acute Urinary Retention (AUR) and required temporary catheterization for a median of 3.5 days (IQR: 3–7). Among these, only 3 (15%) sought emergency room evaluation in the hours/days following the procedure, while the rest were managed in the discharge room in the first hours after the biopsy. Other 43 patients (9.4%) reported mild non-specific voiding symptoms (dysuria) for several days following the procedure, in the absence of AUR. Most of these symptoms were associated with Urinary Tract Infections (UTIs).

UTIs treated with antibiotics occurred in 20 (4.4%) patients. Among them, 18 (90%) were managed at home with oral antibiotic therapy following a telephone consultation with healthcare professionals via a dedicated phone number. The remaining 2 men, due to their condition (high fever > 39 °C, comorbidity requiring clinical monitoring, need for specific antibiotic therapies based on susceptibility testing), were hospitalized for a median of 7 days (IQR: 7–7) to receive intravenous antibiotic therapy. However, it is noteworthy that none of these cases met the criteria for sepsis, and no complications classified as Clavien–Dindo (CD) grade > 2 occurred.

The overall cancer detection rate (CDR) was 63.1% (with 49.5% of clinically significant PCa, defined as ISUP grade group > 2).

At univariable analysis, the only independent predictors of post-procedural generic complications (including bleeding, AUR, and infections) were age at biopsy > 70 years (OR 1.463 [CI: 1.002–2.135], *p* = 0.049) and BMI > 25 kg/m^2^ (OR 1.986 [CI: 1.072–2.68], *p* = 0.029). Age at biopsy > 70 years was also identified as a predictor of post-procedural bleeding (OR 1.578 [CI: 1.071–2.325], *p* = 0.021). At multivariate analysis, only BMI > 25 kg/m^2^ (OR 1.982 [CI: 1.069–3.673], *p* = 0.03) was assessed as a predictor of generic complications after TRPB.

## 4. Discussion

In this audit conducted at a referral high-volume Cancer Centre, we sought to assess the morbidity associated with MRI-fusion-targeted and systematic TRPB. Our findings indicate that TRPB is generally safe, well-tolerated, and associated with a low risk of major adverse events. Of note, our data are at least comparable to those available in the literature, including data from the TP approach. The choice between a transrectal and transperineal approach for MRI-fusion-targeted prostate biopsy remains a topic of debate within the urologic community and across international guidelines [16]. In particular, the European Association of Urology (EAU) guidelines recommend abandoning TRPB in favor of TPPB, due to lower rates of infection and sepsis [13], while the American Urological Association (AUA) recommends either approach, highlighting the absence of prospective RCTs assessing infectious risk [14]. In this scenario, transparent reporting of the morbidity associated with TRPB is crucial to inform patients during shared decision-making as well as to evaluate the cost-effectiveness of different biopsy approaches from clinical and economic perspectives.

In terms of post-procedural outcomes, we observed a 40.7% CD grade ≤ 2 adverse events rates (mostly hematuria and other minor rectal or seminal bleeding), with a very low rate of hospital re-admission (six cases, 1.3%) and no major complications (such as sepsis). While it is important to acknowledge the potential influence of attrition bias (e.g., 8.2% of patients lost to follow-up), our data emphasize that, in experienced hands and with careful preoperative work-up, adverse events following TRPB can be effectively minimized.

While the rate of complications reported in our study aligns with the available literature, it is important to note that, in our study, patients were followed with telephone calls by urologists aiming to assess and manage potential adverse events. This approach may have influenced the rate of emergency room visits, particularly for patients experiencing minor complications (e.g., mild dysuria/hematuria not requiring active treatment).

Supporting our results, in their prospective RCT comparing TRPB and TPPB, Guo et al. [17] reported no difference between TPPB and TRPB considering minor complication rate (44.9% vs. 41.0%, *p* = 0.504), while a significantly higher rate of mild pain in the TPPB group (35.3% vs. 13.0%, *p* < 0.001) was observed. In this study, major complications including high fever (0% vs. 1.2%), sepsis (0% vs. 0.6%), severe rectal bleeding (0% vs. 1.2%), and vasovagal events (0.6% vs. 1.2%) occurred at a significantly lower rate in the TPPB group (0.6%) compared to TRPB (4.3%) (*p* = 0.034). Similarly, the meta-analysis by Xue et al. [18], comparing TRPB and TPPB, showed no significant differences in all complications, including hematuria (20.6% for TRPB vs. 17.1% for TPPB; OR = 1.14, 95% CI = 0.85, 1.53), hematospermia (0.7% for TRPB vs. 1.2% for TPPB; OR = 0.59, 95% CI = 0.14, 2.47), sepsis (0.4% for TRPB vs. 0.4% for TPPB; OR = 0.93, 95% CI = 0.15, 5.82), and AUR (3.8% for TRPB vs. 2.4% for TPPB; OR = 1.39, 95% CI = 0.57, 3.37).

Conversely, a systematic review by Bennett et al., which evaluated 165 articles covering a total of 162,577 patients, reported a higher rate of hospitalization (1.1% vs. 0.9%) and sepsis (0.8% vs. 0.1%) for the TR approach compared with the TP route [19]. Several other studies and reviews [1,20] have sought to determine whether a significant difference in complications exists between these two procedures. However, the results remain uncertain, raising questions about the real impact of the route on complications.

Considering antibiotic prophylaxis for TRPB, in a recent meta-analysis, Pilats et al. [9] reported a rate of infections of 5.6% for patients subjected to prophylaxis vs. 11.6% if no antibiotic prophylaxis was given, reaffirming the importance of adequate preparation to minimize potential complications. While our study reported a slightly lower rate of infectious complications (4.4%) despite antibiotic prophylaxis, it highlights that other factors, such as Povidone-Iodine disinfection and cleansing enema, may also play a marginal role in mitigating these complications. In this context, rectal cleansing with Povidone-Iodine emerges as a safe, well-tolerated, cost-effective intervention, easily performed with minimal additional time, and has shown positive results in reducing post-TRPB infections, as supported by other studies [5]. While the necessity of antibiotic prophylaxis is unquestionable, ongoing debate surrounds the choice of antibiotics. Numerous studies aim to identify the most effective antibiotic for prostate biopsy, considering local bacterial flora, as well as the pharmacokinetics and pharmacodynamics of specific drugs. Fluoroquinolones and Fosfomycin have been the most investigated classes of antibiotics due to their characteristics, with the majority of studies supporting the use of FMT, which has demonstrated a significant reduction in infectious complications compared to Fluoroquinolones [9,21,22,23]. Supporting these findings, Cai et al. [24] highlighted the potential of FMT in preventing post-TRPB febrile UTI in a cohort of over 1000 patients. The results showed statistically significant differences in favor of the group treated with prophylactic FMT, with rates of symptomatic infections and sepsis at 1.2% vs. 12.9% for UTIs and 0.3% vs. 1.8% for sepsis, respectively. Due to the growing increase in Fluoroquinolone resistance and the decision of the European Commission to ban Fluoroquinolones from antibiotic prophylaxis in urological surgery [7], alternatives are necessary. Considering our results, we confirm that FMT could be considered the antibiotic of choice for the prophylaxis of post-TRPB infections.

The results of our study also confirm the efficacy of TRPB in PCa detecting, showing an overall CDR of 63.1% (49.5% of csPCa), which is at least comparable to previous studies both considering TRPB and TPPB [25,26]. To increase clinically significant PCa detection rates, as well as to avoid unnecessary prostate biopsies and thus the corresponding complications, performing mpMRI diagnostic before the procedure and using a combined approach of systematic and fusion-targeted biopsy is now mandatory, as confirmed by several pieces of evidence [13,27,28,29].

Despite the generally safe morbidity profile of TRPB in our study, an internal audit conducted at our Center in December 2021 led to a gradual transition from the TRPB to the TPPB approach in alignment with the recommendations of the EAU Guidelines [13] (Figure 4).

Since its introduction to our center in 2022, TPPB has primarily been reserved for patients deemed at a higher risk of developing infectious complications. It was also chosen for individuals with lesions identified at the apical portion or apex of the prostate gland through mpMRI, supported by literature indicating superior diagnostic accuracy for this procedure compared to TRPB in cases of tumors located in these specific areas [30], despite some controversy in the results [25,26]. As our confidence and expertise in this technique have grown over time and with the aim of aligning with European standards, we have progressively broadened the scope of TPPB, expanding its indications and overlapping with those previously associated with TRPB.

Considering the average time required for outpatient TPPB under local anesthesia, which is generally slightly longer compared to TRPB (45 min vs. 30 min in experienced hands) and recognizing the imperative to sustain a substantial number of procedures (over 500 per year) given the high volume of patients with suspected prostate cancer managed at our center, we find ourselves in a transitional phase. As of now, we have not completely abandoned TRPB. Nevertheless, since the onset of 2022, there has been a consistent uptrend in the number of TPPBs performed. This trajectory is expected to lead to a complete replacement of TRPB in the near future, attributed to enhancements in the learning curve and standardization of the procedure.

Our findings should be interpreted in light of certain study limitations. Firstly, despite our efforts to provide a comprehensive account of the real-life morbidity associated with TRPB, it is important to acknowledge the potential influence of attrition bias and residual confounding on our results. This is particularly pertinent as adverse events were assessed through telephone calls, relying on patients’ self-reported incidences and their management. Consequently, there is a theoretical possibility that the overall morbidity of TRPB in our study might have been underestimated. It is noteworthy, however, that major adverse events necessitating hospitalization within 30 days after the procedure would likely have been managed at our tertiary referral center. Secondly, we acknowledge the possibility that we might not have captured all potential predictors of post-operative complications among routinely collected patient baseline data. Additionally, it is important to consider the heterogeneity among the urologists who performed the procedures. Despite their collective experience in performing such procedures, the diversity in individual approaches could introduce bias in the evaluation of results. Lastly, our study did not extend the assessment of the risk of post-procedural adverse events beyond the 30-day timeframe.

Acknowledging these limitations, our study affirms that TRPB can be a safe procedure when conducted by experienced practitioners and with a standardized procedural work-up. This understanding allows for a smooth transition to a transperineal approach, a shift that can be tailored to the specific resources and clinical needs of the hospital.

Further randomized studies are necessary to more comprehensively compare TRPB with TPPB. This is essential to provide a more precise answer to the ongoing debate surrounding the strengths and limitations of these two techniques. Such studies will play a crucial role in informing patients during shared decision-making and aiding healthcare professionals in choosing the most tailored strategy for each individual case.

## 5. Conclusions

Our study underscores the importance of a standardized, structured, multistep pathway in minimizing the morbidity of TRPB. While a transition to the transperineal approach is recommended and desirable, it is noteworthy that TRPB remains a safe and cost-effective procedure with a low risk of severe adverse events when performed by experienced practitioners following a standardized pathway.

Further studies are needed to pinpoint the specific clinical scenarios in which TRPB could serve as a valid alternative to TPPB, taking into account both clinical considerations and logistical factors.

## Figures and Tables

**Figure 1 cancers-15-05798-f001:**
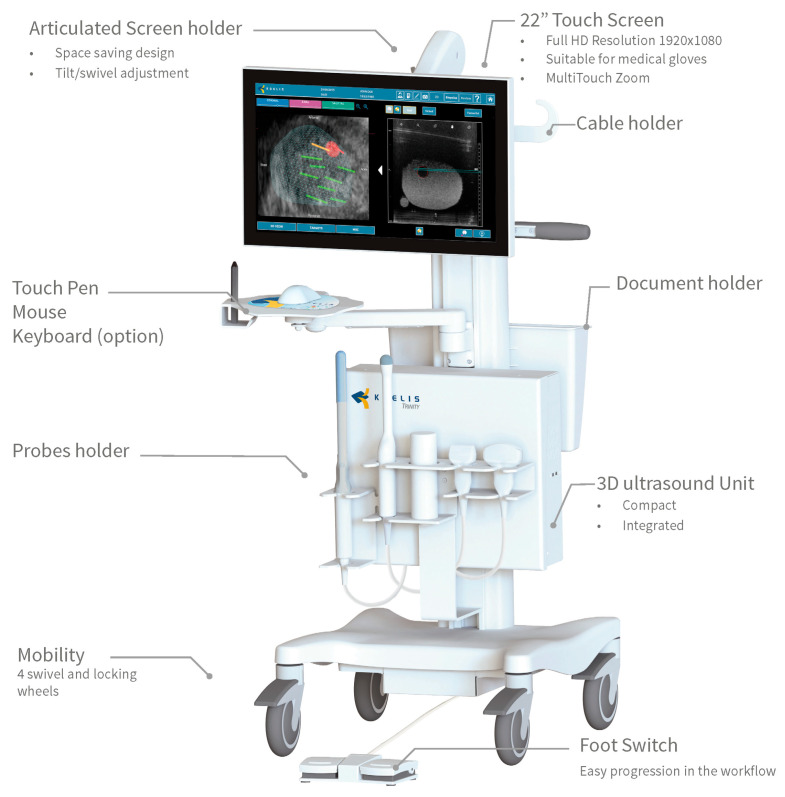
KOELIS Trinity^®^ MRI TPUS Biopsy System.

**Figure 2 cancers-15-05798-f002:**
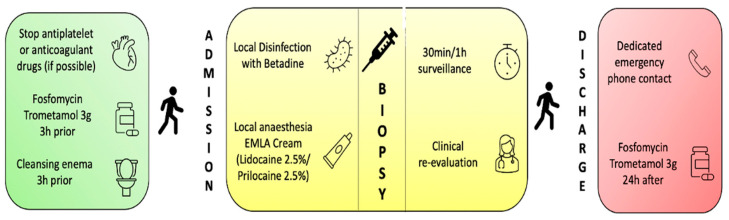
Standardized structured multistep pathway for TRPB.

**Figure 3 cancers-15-05798-f003:**
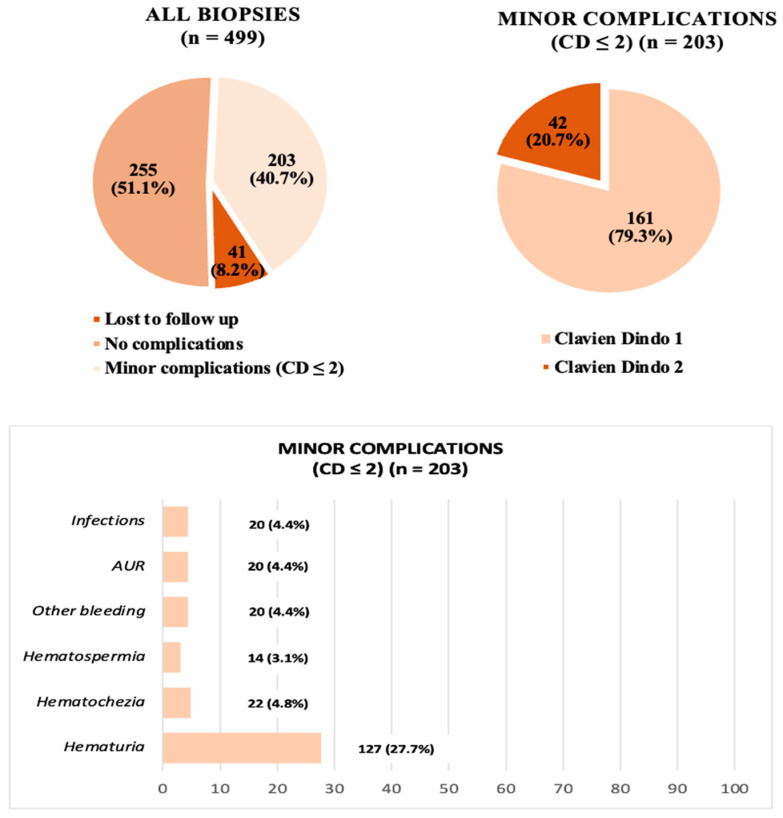
Complications after TRPB.

**Figure 4 cancers-15-05798-f004:**
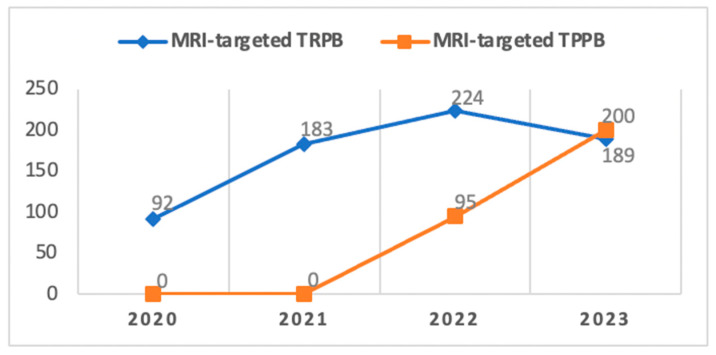
Transition over time at our institution from TRPB to TPPB.

**Table 1 cancers-15-05798-t001:** Baseline characteristics.

*Characteristics*	*TRPB (n = 499)*
**Age** [years], *median (IQR)*	67 (61–73)
**PSA** [ng/mL], *median (IQR)*	6.4 (4.9–9)
**Free/Tot PSA** [%], *median (IQR)*	15 (10.9–19)
**PSA density** [ng/mL^2^], *median (IQR)*	0.12 (0.08–0.18)
**Positive/suspicious DRE**, *n (%)*	252/487 (51.7%)
**CCI** (not age-adjusted), *median (IQR)*	2 (1–3)
**Comorbidities:** - Diabetes mellitus, *n (%)* - CD (not hypertension), *n (%)* - Hemorrhoids, *n (%)*	35/347 (8%) 42/433 (9.7%) 51/346 (14.7%)
**Antiplatelet/anticoagulant drug**, *n (%)* - Suspended before TRPB, *n (%)* - NOT suspended before TRPB, *n (%)*	83/430 (19.3%) - 46/83 (55.4%) - 37/83 (44.6%)
**BMI** [kg/m^2^], *median (IQR)*	25.7 (23.4–27.8)

DRE = Digital Rectal Examination; CCI = Charlson Comorbidity Index; CD = Cardiovascular Disease; BMI = Body Mass Index.

## Data Availability

Correspondence and requests for materials should be addressed to the authors.

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
