# Peer review of "Morbidity of Transrectal MRI-Fusion Targeted Prostate Biopsy at a Tertiary Referral Academic Centre: An Audit to Guide the Transition to the Transperineal Approach"

_cancers, 2023, doi:10.3390/cancers15245798_

Round 1
Reviewer 1 Report
Comments and Suggestions for Authors
I enjoyed reviewing this manuscript, it regards a straightforward analysis of a tertiary center diagnostic transrectal biopsy workflow and its outcomes regarding complications. The authors report a slightly lower infection rate compared to the literature, and show that transrectal biopsy is still a safe procedure with very low likelihood of serious complications; when performed in an experienced high-volume center.
In its present form, it does not contain much novel information for the reader.
I think the manuscript can be further improved if the following is added:
1. Please provide more information regarding selection of patients for transrectal biopsy; is there any pattern of selection noticeable during the transition to transperineal pathway? E.g. regarding selection of PI-RADS lesions or location of the lesions?
2. Are biopsies standardly performed systematic AND target biopsy? Or also target biopsy alone? Is there any correlation between number of cores taken and risk of infection?
3. Is data regarding experience of the urologist available and does this correlate to risk of infection?
4. In how many of the cases was additional biopsy needed (transperineally), of all transrectal biopsies taken? This is relevant information and shows the potential benefit of transperineal biopsy.
5. Addition of complication rates of the transperineal prostate biopsies performed will improve the manuscript and novelty; and could show a decrease in complications over tim
Minor comments:
The English wording can be improve in some cases: e.g. sentence 70 “leaves many logistic challenges behind”, please check to ensure adequate academic English
Comments on the Quality of English Language-
Author Response
- Please provide more information regarding selection of patients for transrectal biopsy; is there any pattern of selection noticeable during the transition to transperineal pathway? E.g. regarding selection of PI-RADS lesions or location of the lesions?
Thank you for the comment. While until 2022 transrectal biopsy was used in all patients with suspected PCa, after the introduction of the transperineal technique we initially used this procedure for patients considered at high infectious risk, or with mpMRI-identified lesions at the level of the prostatic apex or the anterior portion of the gland, for which there is evidence in the literature reporting a superiority of the transperineal approach in terms of detection rate. Subsequently, thanks also to greater manageability and experience acquired over time, we have expanded these indications to the point of superimposing this procedure with the transrectal technique, regardless of the location of the lesions. We added a short paragraph in the discussion to explain this transition process (lines 278-286).
- Are biopsies standardly performed systematic AND target biopsy? Or also target biopsy alone? Is there any correlation between number of cores taken and risk of infection?
Thanks for your request for clarification. In our series, all biopsies were standardly performed taking both target and systematic cores, as recommended by the guidelines. As reported in the results section, median number of biopsy cores taken during the procedure was 16 (IQR: 14 – 18), with a median of respectively 4 (IQR: 4 – 6) and 11 (IQR: 10 – 12) cores for target and random samplings. However, we have inserted a sentence to clarify this concept in the materials and methods section (line 112), as it was perhaps not clearly specified previously.
As regards a correlation between number of cores taken and risk of infection, our analyzes did not reveal any statistically significant significance in this regard.
- Is data regarding experience of the urologist available and does this correlate to risk of infection?
The biopsies in this series were performed by a team of urologists in our center with several years of experience, however there is no data available on who among them personally performed each procedure, in order to correlate it with the actual risk. This is certainly a limitation of our study, as this heterogeneity could have a confounding role and cause bias. We brought this back into the discussion, adding it to the limitations of the study.
- In how many of the cases was additional biopsy needed (transperineally), of all transrectal biopsies taken? This is relevant information and shows the potential benefit of transperineal biopsy.
Thank you for the question. Since the aim of our study was to evaluate post-procedural complications, which typically emerge early after the procedure, rather than focusing on the detection rate and diagnostic accuracy of these procedures, we do not have available standardized follow-up data over time for all these patients, unlike the data on how many of these had already undergone a prostate biopsy in the past, as it was recorded at the time of medical history collection.
However, considering those who have received a negative histological result for cancer (as well as those who have been included in an active surveillance process), for whom the possibility of a further biopsy could be envisaged, also in consideration of the fact that this is a rather recent cohort of patients, so it cannot be ruled out that in the future some of them may require a new biopsy, in addition to the fact that we only started performing transperineal biopsies just under 2 years ago, considering the data at this moment at our disposal only a very small percentage (about 1.5%) have undergone a new biopsy (transrectal or transperineal), so we think we do not have the numbers to make these considerations.
- Addition of complication rates of the transperineal prostate biopsies performed will improve the manuscript and novelty; and could show a decrease in complications over time
We completely agree with your consideration, and we really appreciate the suggestion. We are currently working to collect this data and then propose a new paper that can compare these two procedures. However, since our work aimed to focus mainly on the transrectal technique, in order to prove that it can still be a card to play in the diagnosis of prostate cancer, taking the right precautions and adhering to the best possible practices, we did not consider reporting data on transperineal biopsies, which will certainly be a source of further studies. We would therefore like to continue with the practice of transperineal biopsies and then present our results with a more adequate number, since at the moment our data in this regard are still quite limited
Minor comments:
The English wording can be improve in some cases: e.g. sentence 70 “leaves many logistic challenges behind”, please check to ensure adequate academic English
Thanks for the comment. We have tried to improve the form and level of English, making the text more fluent, as requested
Reviewer 2 Report
Comments and Suggestions for Authors
One of the major arguments in favor of TP biopsy is the theoretical low infection risk - however recent studies start to demonstrate that in reality this perceived benefit might not hold true. I am glad to see the manuscript by Alberti et.al. showed that using a standard protocol including the utilization of pre-procedural antibiotic prophylaxis, infection was actually not the primary complications following TP biopsy. The results presented in this manuscript definitely encouraged the urological community to reconsider the alleged benefits of TP biopsy.
Author Response
Thank you for your support and comments. In this work we tried to report our experience and our standardized pathway for transrectal biopsies, so that clinical-surgical practice can be improved from the exchange of knowledge. Although the debate on which procedure can be considered the best is always ongoing, there seems to be a predominance of opinions in favor of the transperineal technique, also considering the strong recommendations of the European guidelines. This, however, should not be considered as a dogma, but we should reflect on whether there really is one technique that is better than the other in an absolute sense, or whether rather each of them may have strengths and limitations to take into account, trying to use them as best as possible.
Reviewer 3 Report
Comments and Suggestions for Authors
The study investigates the morbidity associated with transrectal prostate biopsy (TRPB) at a tertiary referral Academic Centre, with a focus on the impact of a standardized clinical pathway. The objective is to provide critical insights into the safety and effectiveness of TRPB compared to the transperineal approach. The authors should be congratulated for the work and for providing a new topic to the literature. The manuscript is interesting and easy to read. In my opinion, it is suitable for publication, after major revision.
MAJOR COMMENTS
- Introduction. Strengthen the introduction by emphasizing the global debate on the choice between transrectal and transperineal biopsy approaches. Highlight the relevance of the study in this context.
- Material and Methods. Provide more information on the patient selection criteria and inclusion/exclusion criteria.
- Discussion. Strengthen the discussion by comparing your findings with existing literature on TRPB and TPPB. Stresses the role of MRI in the detection of prostate cancer , as reported in these articles (doi: 10.1016/j.clgc.2022.04.013, 10.1007/s00261-020-02798-8).
- Discuss the limitations of the study thoroughly, including potential biases and areas for future research.
- Emphasize the clinical implications of your results and how they contribute to the ongoing debate between TRPB and TPPB.
Comments on the Quality of English LanguageMinor editing
Author Response
Thank you for your comments on our article, we are grateful for the opportunity to revise the manuscript and fix some unclear points.
We have tried to improve the various sections as proposed and included the suggested references as a starting point to improve the discussion. We also tried to improve the form and level of English, making the text more fluent.
We hope our new version will be appreciated.
Round 2
Reviewer 1 Report
Comments and Suggestions for Authors
the comments are satisfactory, accept for publication
Reviewer 3 Report
Comments and Suggestions for Authors
Authors answered all comments and suggestions.
Comments on the Quality of English LanguageMinor editing.